# Mitochondrial Retinopathies

**DOI:** 10.3390/ijms23010210

**Published:** 2021-12-25

**Authors:** Massimo Zeviani, Valerio Carelli

**Affiliations:** 1Department of Neurosciences, The Clinical School, University of Padova, 35128 Padova, Italy; 2Veneto Institute of Molecular Medicine, Via Orus 2, 35128 Padova, Italy; 3Department of Biomedical and Neuromotor Sciences, University of Bologna, 40139 Bologna, Italy; 4Programma di Neurogenetica, IRCCS Istituto delle Scienze Neurologiche di Bologna, Via Altura 6, 40139 Bologna, Italy

**Keywords:** retina, mitochondrial disorders, mitochondrial DNA, retinitis pigmentosa, optic atrophy, mtDNA heteroplasmic deletions, Kearns-Sayre syndrome, neurogenic muscle weakness, ataxia and retinitis pigmentosa (NARP), Leber’s hereditary optic neuropathy (LHON), autosomal dominant optic atrophy (ADOA)

## Abstract

The retina is an exquisite target for defects of oxidative phosphorylation (OXPHOS) associated with mitochondrial impairment. Retinal involvement occurs in two ways, retinal dystrophy (retinitis pigmentosa) and subacute or chronic optic atrophy, which are the most common clinical entities. Both can present as isolated or virtually exclusive conditions, or as part of more complex, frequently multisystem syndromes. In most cases, mutations of mtDNA have been found in association with mitochondrial retinopathy. The main genetic abnormalities of mtDNA include mutations associated with neurogenic muscle weakness, ataxia and retinitis pigmentosa (NARP) sometimes with earlier onset and increased severity (maternally inherited Leigh syndrome, MILS), single large-scale deletions determining Kearns–Sayre syndrome (KSS, of which retinal dystrophy is a cardinal symptom), and mutations, particularly in mtDNA-encoded ND genes, associated with Leber hereditary optic neuropathy (LHON). However, mutations in nuclear genes can also cause mitochondrial retinopathy, including autosomal recessive phenocopies of LHON, and slowly progressive optic atrophy caused by dominant or, more rarely, recessive, mutations in the fusion/mitochondrial shaping protein OPA1, encoded by a nuclear gene on chromosome 3q29.

## 1. Mitochondrial Bioenergetics

This review is focused on retinopathy caused by primary genetic mitochondrial disorders, although mitochondrial dysfunction may be an important pathogenetic component of other retinopathies. Present in virtually all eukaryotes, mitochondria are double-membraned organelles with a central role as the powerhouses of the cell [1]. They are in fact the major source of the high-energy phosphate molecule, adenosine triphosphate (ATP), synthesized by the mitochondrial respiratory chain (MRC) through the process of oxidative phosphorylation (OXPHOS) [2]. ATP is required for all active processes in the cell, and ATP deficiency leads to cellular dysfunction and ultimately cell death. ATP is synthesized by mitochondrial oligomycin-sensitive, proton (H^+^)-dependent ATP synthase, (also termed as complex V, cV), which is a multi-subunit structure embedded in the IMM and strictly linked to the MRC. Since cV can also act in reverse mode, i.e., as an ATPase which hydrolyses ATP into ADP + Pi, and this is in fact the usual way it is measured spectrophotometrically, it is also called oligomycin-sensitive mitochondrial ATPase. The MRC and the H^+^-ATPase/synthase stores the energy liberated through respiration, i.e., the stepwise flow of electrons extracted by oxidative catabolism from nutrient-derived substrates, as ATP and transfers it through the MRC complexes cI or cII, to cIII and eventually to cIV, due to mobile electron shuttles, Coenzyme Q (from cI-cII to cIII) and cytochrome c (from cIII to cIV). In turn, cIV (cytochrome c oxidase, COX), fixates the electrons to molecular O_2_, the final “electron sink” of the pathway, converting it into H_2_O. The energy liberated by this electronic current is exploited by cI, cIII, and cIV to pump protons across the inner mitochondrial membrane (IMM). As a result, energy is conserved generating a mitochondrial membrane potential, MtMP, which is composed of a proton concentration gradient (∆pH) and an electrostatic gradient, more negative outside, but more positive inside, the IMM (∆Ψ). The MtMP provides the proton-motive force (∆*p*) [3,4] that drives the rotational engine of H^+^-ATP synthase, by exploiting the protons flowing from the outside to the inside of the IMM through the subunit A channel, to ultimately cause the condensation of ADP and Pi into ATP. In addition to ATP production through OXPHOS, ∆*p* can also be dissipated into heat, a crucial function, especially in homeothermic animals, which is controlled by uncoupling proteins [5]. In addition, the MtMP provides energy for numerous other processes, such as trafficking of Ca^2+^, Fe^2+^, and other important ions, reactive oxygen and nitrous oxygen species (ROS and NOS) production, mitochondrial protein translocation, etc. A host of other pathways residing within mitochondria can modify and influence OXPHOS, and OXPHOS abnormalities can in turn generate signals triggering either homeostatic pathways, e.g., mitochondrial biogenesis, or execution programs, e.g., mitophagy, which eliminates some impaired parts, or the whole spent organelle, or apoptosis, which eliminates the dysfunctional cell altogether [6]. This complex metabolic network is largely based on the repertoire of the entire mitochondrial proteome, summing up to approximately 1500 proteins in mammals [7]. Mitochondria contain their own DNA (mitochondrial DNA, mtDNA), which in mammals is a 16.5 kb circular molecule of double-stranded DNA that encodes 13 polypeptides and 24 RNAs (22 transfer tRNAs and two ribosomal rRNAs), essential for in situ protein synthesis. This is also needed because the genetic code of mtDNA differs in different phyla, including vertebrates, from the universal code, which makes the two genomes, nuclear and mitochondrial, reciprocally untranslatable. MtDNA polypeptides are all subunits of four OXPHOS complexes (c), namely cI (*n*. 7), cIII (*n*. 1), cIV (*n*. 3) and cV (*n*. 2), where they interact with over 70 nucleus-encoded protein subunits and several non-protein prosthetic groups [8]. Additional nucleus-encoded factors are essential for mtDNA maintenance and expression, as well as for MRC formation and activity.

In contrast with mitochondrion-related nuclear DNA (nDNA) genes, present in a diploid organization in somatic cells, and transmitted by mendelian inheritance, mtDNA is present in thousands of copies in each cell, (polyplasmy) and, in sexuate organisms, its transmission from one generation to the next occurs exclusively through the female gamete. Uniparental maternal inheritance is also the way transmissible deleterious mtDNA mutations, leading to disease, are passed to the offspring. As a result, genetic defects affecting mtDNA or OXPHOS-related nDNA genes can impair ATP synthesis, determine mitochondrial dysfunction, and cause human disease [9]. Figure 1 depicts the major mtDNA-related syndromes against a scheme of the molecule.

Mitochondrial disorders are in fact clinical entities associated with the ascertained or allegedly genetic defects of mitochondrial OXPHOS [10]. Tissue and organ functions where energy demand is high, such as neurons and muscle fibres, critically depend on adequate ATP supply. This explains why primary mitochondrial disorders usually cause neurodegeneration and/or muscle weakness, in children and adults [11]. However, specific mitochondrial syndromes may involve any other organ, either individually or in combination with brain and muscle dysfunction. The double genetic contribution and the complexity of the OXPHOS-related biochemical network account for the extreme clinical and genetic heterogeneity of mitochondrial disorders. In addition, many pathogenic mtDNA mutations are heteroplasmic, that is, they co-exist with a variable percentage of wild-type (wt) mtDNA, and this proportion can differ from tissue to tissue and in each individual [12], partly dictating the clinical outcome and the tissue-specific involvement in different subjects. Because of polyplasmy, the transmission of mutant vs. wt-mtDNA is largely dependent on the stochastic distribution of the organelles during mitosis in somatic cells, as well as meiotic divisions in female germ cells, and on the capacity of different tissues to work out effective selection against OXPHOS-defective cells. Thus, the same mutation, for instance the 3243G > A transition in the gene encoding mt-tRNA^Leu(UUR)^, can be associated with a virtually disease-free individual when the percentage of heteroplasmy in critical tissues is low or cause a range of different clinical syndromes, from relatively benign hearing loss and type 2 diabetes, to late-onset myopathy, to a devastating juvenile or infantile syndrome known as mitochondrial encephalomyopathy with lactic acidosis and stroke-like episodes (MELAS) [13]. Although individually rare, when taken as a group primary mitochondrial disorders are a frequent category of monogenic diseases (~1 affected individual in 4300 live births in Europe) [14]. However, the number of individuals carrying a mtDNA mutation in a percentage below the clinical threshold is probably much higher, around 1 in 500 live births. 

## 2. Retina Is a Preferential Target of Mitochondrial Dysfunction and Diseases

Retinal dystrophy with features resembling retinitis pigmentosa and maculopathies, as well as optic neuropathy with retinal ganglion cells (RGCs) loss and atrophy of the optic nerve are very frequent in human mitochondrial disorders [15]. 

## 3. Functional Anatomy of the Retina

The retina is an extra-encephalic extension of the central nervous system (CNS) [16], composed of light (photo)receptors, rods, cones and a subset of retinal ganglion cells (RGCs) expressing the photopigment melanopsin (mRGCs) [17], which establish synaptic connections with several neuronal cells, eventually converging on the RGC layer as the final output projecting to the brain and ultimately the occipital cortex, where vision is formed. In fact, from RGCs, unmyelinated axons form the retinal fibre layer gathering on the optic disc, eventually forming the myelinated optic nerve, once they have passed the lamina cribrosa. Most axons through the chiasm and optic tracts reach the lateral geniculatus nucleus (LGN) serving the image-forming visual pathway; others project to the suprachiasmatic nucleus of the hypothalamus serving the non-imaging forming function, mainly deputed to photoentraining the circadian rhythms. As shown in Figure 2A, from the outer to the inner areas, the retina is composed of ten layers: (1) retinal pigmented epithelium (RPE), (2) photoreceptors layer, (3) outer limiting membrane, (4) outer nuclear layer containing the cell bodies of rods and cones, (5) outer plexiform layer composed of synapses between dendrites of horizontal cells from the inner nuclear layer and photoreceptor cells, (6) inner nuclear layer with cell bodies of horizontal cells, bipolar cells, amacrine cells, interplexiform neurons, Müller cells, (7) inner plexiform layer composed of synapses between axons of bipolar cells and dendrites of RGCs, (8) retinal ganglion cells (RGCs plus mRGCs photoreceptors) layer, (9) retinal nerve fibres layer (RNFL), and (10) inner limiting membrane. The RPE is in tight functional conjunction with photoreceptors, i.e., *cones* responsible for colour vision, and *rods*, deputed to detect dim light and contrast sensitivity serving central vision, which transduce light into action potentials, then elaborated by the retinal circuitry to convey visual information to the RGCs axons ultimately forming the optic nerve.

## 4. Physiology of the Photoreceptors

As light passes through the lens and the vitreous, it hits the retina from the inside of the eye, thus penetrating through the entire retinal thickness to eventually reach rods and cones at the outer edge. In the central foveal region of the retina, the inside layers are pulled aside from the fovea, a minute area in the centre of the retina, occupying a total surface of 1 mm^2^, deputed to provide visual acuity and detailed vision. The central fovea, only 0.3 mm in diameter, is composed almost entirely of slender cones. Additionally, the blood vessels, RGCs, inner nuclear layer, and plexiform layers are all displaced on the sides, letting light reach unimpeded the cones. The surrounding macular area is the on-ly region of the retina characterized by multilayered RGCs, functional to provide the highest visual discrimination, the so-called visual acuity.

The major functional segments of either a rod or cone are: (1) the outer segment, conical in shape in the cone, rod-like in the rod, (2) the inner segment, (3) the nucleus, and (4) the synaptic body. The light-sensitive photochemical is found in the outer segment. In the rods is rhodopsin; in the cones there is one of three colour pigments, functioning almost exactly the same as rhodopsin except for differences in spectral sensitivity.

Each outer segment of rods and cones are packed with approximately 1000 piled up discs, each being an infolded shelf of cell membrane. Both rhodopsin and the colour pigments are incorporated into the membranes of the discs. The pigments constitute about 40 percent of the entire mass of the outer segment. The *inner segment* of the rod or cone contains cytoplasmic organelles, especially mitochondria, which provide energy. The synaptic body is the portion of the rod or cone that connects with subsequent neuronal cells, the horizontal and bipolar cells, the next stages in the retinal circuitry.

When exposed to light, the resulting photoreceptor potential causes increased negativity of the membrane potential, which is a state of hyperpolarization. This is exactly the opposite to the decreased negativity (the process of “depolarization”) that occurs in almost all other sensory receptors. When rhodopsin is decomposed by light, it decreases the rod membrane conductance for Na^+^ ions in the outer segment of the rod, resulting in the hyperpolarization of the entire rod membrane. In the light cGMP accumulates and the cGMP-gated Na^+^ channels are closed, thus reducing the inward Na^+^ current. Na^+^ ions are continuously pumped outward through the membrane of the inner segment. Thus, more sodium ions now leave the rod than leak back in, creating increased negativity inside the membrane, in proportion to the amount of light energy striking the rod. The photochemicals in the cones have almost exactly the same chemical composition as that of rhodopsin in the rods. The only difference is that the protein portions, or cone-specific photopsins, are slightly different from the scotopsin of the rods. The retinal portion of all the visual pigments is exactly the same in the cones as in the rods. The colour-sensitive pigments of the cones, therefore, are combinations of retinal and photopsins. Only one of three types of colour pigments is present in each cone, thus making the cones selectively sensitive to different colours: blue, green, or red, with peak absorbencies at 445, 535, and 570 nm, respectively. 

The different neuronal cell types of the retina are as follows: The photoreceptors—rods and cones—which transmit signals to the outer plexiform layer, where they synapse with bipolar cells and horizontal cells.The horizontal cells, which transmit signals horizontally in the outer plexiform layer from the rods and cones to bipolar cells.The bipolar cells, which transmit signals vertically from the rods, cones, and horizontal cells to the inner plexiform layer, where they synapse with ganglion cells and amacrine cells.The amacrine cells, which transmit signals in two directions, either directly from bipolar cells to RGCs or horizontally within the inner plexiform layer from axons of the bipolar cells to dendrites of the ganglion cells or to other amacrine cells.The ganglion cells, or RGCs, which transmit output signals from the retina through the optic nerve into the brain serving both the visual and non-visual pathways (mRGCs). The first leads to formed vision, whereas the second is instrumental to photoentrain circadian rhythms.

Finally, the interplexiform cell transmits inhibitory signals in the retrograde direction from the inner plexiform layer to the outer plexiform layer, thus controlling lateral spread of visual signals by the horizontal cells in the outer plexiform layer and the degree of contrast in the visual image.

Both rods and cones release *glutamate* at their synapses with the bipolar cells. Amacrine cells secrete at least eight types of neurotransmitters, including γ-aminobutyric acid (GABA), glycine, dopamine, acetylcholine, and indolamine, all of which normally function as inhibitory transmitters. 

Virtually all the retinal neurons, including the photoreceptors and except the RGCs, conduct their visual signals by electrotonic conduction. This allows graded conduction of signal strength directly related to the intensity of illumination. Thus, the signal is not all or none, as would be the case for action potentials. The only retinal neurons operating by action potentials are the RGCs. 

Each retina contains about 100 million rods and 3 million cones, yet the number of RGCs is only about 1.2 million. Thus, an average of 60 rods and 2 cones converge on each ganglion cell and the corresponding optic nerve axon connecting to the brain. As fovea is approached, fewer rods and cones converge on each RGC axon, and the rods and cones also become slender. These effects progressively increase the acuity of vision in the central retina. In the central fovea there are only slender cones—about 35,000 of them—and no rods. Additionally, the number of RGC axons leading from this part of the retina is almost exactly equal to the number of cones. This explains the high degree of visual acuity in the central retina in comparison with peripheral retina. 

The peripheral retina is much more sensitive than the fovea to weak light, which occurs partly because rods are 30 to 300 times more sensitive to light than cones are. In addition, as many as 200 rods converge on a single RGC axon in the more peripheral portions of the retina, and thus signals from the rods summate to give even more intense stimulation of the peripheral RGCs. 

## 5. The Retinal Pigmented Epithelium (RPE)

The RPE has several functions, summarized here.

Light absorption: RPE is responsible for absorbing scattered light, thus improving the quality of the optical system, and protecting it, by melanosomes, from photo-oxidative stress due to a strong concentration of photo-oxidative energy operated by the lens. The high perfusion of the retina creates a high oxygen tension environment. The combination of light and oxygen brings oxidative stress, and RPE has many mechanisms to cope with it.

Epithelial transport: RPE composes the outer blood-retinal barrier, the epithelia have tight junctions between the lateral surfaces providing an isolation of the inner retina. This is important for the immune privilege of eyes, a highly selective transport of substances for a tightly controlled environment. RPE supplies nutrients to photoreceptors, including glucose and ketone bodies, controls ion homeostasis and eliminates water and metabolites.

Spatial buffering of ions: ion changes in the subretinal space are fast and require a capacitive compensation by RPE [17], where voltage-dependent channels operate the transepithelial ion transport [20].

Visual cycle: The visual cycle needs to be adapted to different visual requirements such as vision in darkness or lightness. RPE provides the storage of the photosensitive molecule, “retinal”, which is part of rhodopsin and cone opsins, and the adaption of the reaction speed. In the transition from darkness to light, large amount of 11-cis retinal is required and provided by the RPE. 

Phagocytosis of photoreceptor outer segment (POS) membranes: POS are exposed to constant photo-oxidative stress, and they go through constant destruction by it. They are constantly renewed by shedding their end, which RPE then phagocytoses and digests.

Secretion: The RPE closely interacts with photoreceptors on one side and with cells on the blood side of the epithelium, such as choroid endothelial cells or cells of the immune system. In order to communicate with the neighbouring tissues the RPE is able to secrete a large variety of factors and signalling molecules, such as ATP, fas-ligand (fas-L), fibroblast growth factors (FGF-1, FGF-2, and FGF-5), transforming growth factor-β (TGF-β), insulin-like growth factor-1 (IGF-1), ciliary neurotrophic factor (CNTF), platelet-derived growth factor (PDGF), vascular endothelial growth factor (VEGF), lens epithelium-derived growth factor (LEDGF), members of the interleukin family, tissue inhibitor of matrix metalloproteinase (TIMP) and pigment epithelium-derived factor (PEDF). 

Immune privilege of the eye: The inner eye represents an immune privileged space, which is disconnected from the immune system of the blood stream. This is largely provided by the RPE as a mechanical tight barrier separating the inner space of the eye from the blood stream and by silencing immune reaction in the healthy eye or activating the immune system in disease.

## 6. Bioenergetics of Photoreceptors

Human photoreceptors have a poorly understood, but very high, energy metabolism. Their nutrient supply derives from a layer of capillaries adjacent to the Bruch’s membrane in the choroid, the choriocapillaris, taking part in a “metabolic ecosystem” comprising the RPE and adjacent Müller cells [21]. OXPHOS is an essential source of ATP for the retina [22], especially the mitochondrial-rich inner segments of the photoreceptors [23]. However, aerobic glycolysis is also active (the Warburg effect) [24,25,26,27,28]. Photoreceptors are one of the highest consumers of mitochondrial-generated ATP, and in fact the inner part of their outer segment is a “mitochondrial bag”, sustaining the energy demand, particularly for the external part of the outer segment. The main source of electron donors feeding photoreceptor OXPHOS is the RPE. RPE cells feed receptors with glucose, but also ketone bodies. Mitochondria act also as a signalling platform that communicates changes through retrograde/anterograde cues. Reactive oxygen species (ROS), produced as a by-product of electron transfer act as important signal molecules onto the nucleus, activating mitochondrial biogenesis. Unfolded protein response, ATP and Ca^++^ coordinate communication between mitochondria and nucleus as well. Finally, extensive mitochondrial damage can initiate autophagy of portions of the organelle or of the entire organelle, and eventually cell death via the release of mitochondrial apoptogenic proteins and mtDNA molecules, through opening of the permeability transition pore. This event triggers apoptosis and inflammasome activation, necroptosis or pyroptosis. MtDNA release into the cytoplasm activates the inflammasome and triggers sterile inflammation in distant cells. 

## 7. Physiology of RGCs and Related Axons Comprising the Visual Pathway

The RGC start the visual pathway, running first juxtaposed to the surface of inner retina as retinal nerve fibre layer (RNFL), converging to the optic disc, crossing the lamina cribrosa, exiting in bundles that acquire oligodendrocytic myelin sheet, thus forming the optic nerves and eventually initiating the multi-neuronal optic pathway within the CNS. The visual pathway eventually projects on the fissure-calcarine area of the occipital lobe the primary visual cortex. Remarkably, the RGCs axons are not myelinated until they cross the lamina cribrosa to emerge on the posterior edge of the eyeball (Figure 2B). In the non-myelinated part of RGC axons at RNFL the action potentials travel in a continuous fashion, in contrast with the saltatory conduction occurring in the myelinated portion of the axons, in the retrobulbar optic nerve. Continuous progression of action potentials is slower and requires a much higher amount of energy compared to saltatory conduction. This is why the non-myelinated portion of the optic nerve fibres has many more mitochondria relative to the myelinated portion, where mitochondria are concentrated around the Ranvier’s spaces [29,30]. Since mitochondria along the axons need to be fragmented to travel, they change shape and density of matrix and cristae, as soon as they make the transition from non-myelinated to myelinated axons [31]. The tight dependency on mitochondrial proficiency for transmitting the action potential is one of the major causes for the exquisite susceptibility of the intra-retinal part of the RGC axons to impaired mitochondrial energy function. The small axons of the papillomacular bundle, which serve the central vision key to provide visual acuity, are the most sensitive to energy depletion, as they cope less efficiently with compensatory mechanisms [32]. 

## 8. Retinopathy in Mitochondrial Disease

The retina is a frequent target in mitochondrial disease. Two retinal components are exquisitely sensitive to OXPHOS impairment, (1) the pigmented/photoreceptor layers and (2) the ganglion cell/nerve fibre layers. As a result, the two most frequent conditions associated with mitochondrial bioenergetics impairment is (1) tapeto-retinal degeneration (retinal dystrophy, also known as retinitis pigmentosa), and (2) optic atrophy. Genes, gene products, main ophthalmological abnormalities, associated extra-ocular signs, if any, and references are shown in Table 1.

## 9. Retinal Dystrophy in Mitochondrial Disease

Photoreceptors require large amounts of energy to maintain their resting potentials [33] with cones incurring an even greater energy expenditure than rods [34]. In the face of this relentless energy demand it seems likely that an impairment of energy metabolism would be detrimental to photoreceptor function with serious consequences for vision. Indeed, there is converging evidence that bioenergetic dysfunction is a key pathogenic factor of cone degeneration in retinal dystrophy, also known as retinitis pigmentosa (RP) [35,36,37,38]. In the majority of subtypes of RP, the genetic defect is expressed in the rods, but in most individuals the cones eventually degenerate resulting in loss of central vision. Studies in animal models of RP have demonstrated that high glucose is critical for cone survival and that reduced glucose entry into cones triggers their degeneration [35,36,37,38]. Moreover, a single injection of glucose has been shown to cause a short-term improvement in cone morphology in a slow-progressing porcine model of RP [38]. The vitreal glucose level protects RGCs against experimental ischemic injury and temporarily recovers contrast sensitivity in individuals with glaucoma [39,40].

RP is frequently associated with some deleterious genetic mutations, most commonly in mtDNA, impairing mitochondrial OXPHOS. Two are the most invariant RP-associated conditions: maternally inherited neurogenic muscle weakness with retinitis pigmentosa (NARP), and sporadic Kearns–Sayre syndrome (KSS). These syndromes are due to (1) maternally inherited heteroplasmic mutations in the ATPase6-encoding mtDNA gene, and (2) sporadic heteroplasmic single deletions of mtDNA, respectively.

## 10. Neurogenic Muscle Weakness, Ataxia and Retinitis Pigmentosa (NARP)

As shown in Table 1. NARP is usually associated with a heteroplasmic transversion in the mtDNA ATPase6 gene (m.8993T > G) [41]. A much rarer m.8993T > C transition has been reported in several families and determines a less severe disease than the m.8993T > G transversion, albeit with very similar if not identical clinical features. For both mutations, the degree of heteroplasmy in critical tissues dictates the severity of the syndrome. When occurring in infancy with high degree of heteroplasmy, the 8993T > G mutation determines a very severe encephalopathy, termed maternally inherited Leigh syndrome (MILS) [42]. In adult-onset NARP, as well as in MILS, RP is an invariable sign, in many cases preceding the onset of ataxia and other motor signs (Figure 3). For patients, progressive evolution from hemeralopia, to tunnel vision, to blindness is often the most disabling symptom and a major source of stress and complaint. The fundus shows the presence of the typical mottled hypopigmented and hyperpigmented areas characterizing RP. Heteroplasmic m.8993T > G values above 70% are usually associated with MILS, whereas <60–70% heteroplasmy is more often associated with adult-onset NARP, with a 15% interval between 70% and 85% determining syndromes of intermediate severity, and >85% usually associated with MILS [43]. In general, heteroplasmic values below 40% determine either no symptom or mild symptoms occurring in advanced age. In contrast with other pathogenic heteroplasmic point mutations of mtDNA, e.g., the m.3243A > G MELAS mutation, the heteroplasmic load of the NARP m.8993T > G mutation, as well as of other rarer NARP/MILS associated mutations in the ATPase6 gene, is similar in different organs, including post-mitotic highly specialized tissues such as skeletal muscle, cardiac muscle and brain, as well as tissues that can be obtained by non-invasive or minimally invasive means, such as blood, urinary or buccal exfoliated cells, hair follicles, etc. This is also true for DNA extracted from chorionic villi samples (CVS), which makes it possible to perform reliable prenatal diagnosis. Whilst the association between the m.8993T > G mutation and NARP/MILS offers a conclusive diagnostic genetic tool, the biochemical evaluation of oligomycin-sensitive ATPase activity is not a reliable diagnostic approach, since this spectrophotometrically detected reaction measures the hydrolytic ATPase proficiency of the enzyme, a function which is largely independent from the role in proton flow played by the protein encoded by the ATPase6 mtDNA gene. As shown in Figure 4, the ATPase6 protein forms in fact an oblique transmembrane channel through which protons pass from the outer to the inner mitochondrial compartments along the electrochemical gradient generated by mitochondrial respiration, thus dissipating it. The m.8993T > G mutation results in the replacement of the highly conserved Leu_156_ to Arg (Leu156Arg) of the ATPase6-encoded proton channel. The replacement of a neutral amino acid (Leu_156_) residue with a positively charged basic residue (Arg) in the proton channel, which is part of the F_0_ ATPase particle, is likely to interfere with the exit of protons promoted by the neighbouring Arg_159_, as the final step of the proton jumping mechanism through hydrogen-bonded water molecules (known as Grotthuss mechanism) occurring in the channel [44]. The mutant Arg_156_ ultimately impairs the proton-motive force, which drives the rotational-based condensation of ADP and Pi into ATP, operated by the F1 particle of the mitochondrial H^+^-ATPase. In addition, the activation of the negative feed-back mechanism linking H^+^-ATPase activity with respiration causes the backlogging of the electron flow along the MRC and the consequent increase in electron leakage, ROS production and oxidative damage to cell structures. The m.8993T > G mutation is also hypothesized to undermine the synthesis of citrulline by carbamoyl-phosphate synthetase 1 (CPS1), and hypocitrullinemia is reported in MILS [45].

## 11. Large-Scale Rearrangements of mtDNA 

Table 1 lists the large-scale rearrangements of mtDNA associated with Retinal degeneration. Figure 5 illustrates the main molecular and clinical features associated with large-scale rearrangements of mtDNA. Single, large-scale rearrangements of mtDNA can be either single partial deletions, or partial duplications. Rearranged molecules, lacking a portion of the mitochondrial genome, can be detected as an independent mtDNA species (single mtDNA deletion, ∆-mtDNA) or joined to a wild-type molecule in a 1:1 ratio, as partially duplicated mtDNA. Frequently, a mixture of the two rearrangements co-exists in the same cell or tissue [48,49]. Three main clinical phenotypes are associated with these mutations: Kearns–Sayre syndrome (KSS), sporadic progressive external ophthalmoplegia (PEO) and Pearson’s syndrome. KSS is a (usually) sporadic disorder characterized by the triad of: (i) chronic progressive external ophthalmoplegia (CPEO); (ii) onset before age of 20 years; and (iii) pigmentary retinopathy (RP). Cerebellar syndrome, heart block, increased CSF protein content, diabetes and short stature are also part of the syndrome. Patients with this disease invariably show RRFs in muscle biopsy [48]. KSS is characterized by neuroradiological abnormalities affecting the deep structures of the brain and the subcortical white matter [50]. Single deletions/duplications can also result in milder phenotypes such as adult-onset CPEO, characterized by late-onset progressive external ophthalmoplegia, proximal myopathy and exercise intolerance. RP is rarely reported in CPEO. In both KSS and CPEO, diabetes mellitus and hearing loss are frequent additional features that may occasionally precede by years the onset of neuromuscular symptoms [51]. Finally, large-scale single deletions/duplications of mtDNA may cause Pearson’s bone-marrow-pancreas syndrome, a rare disorder of early infancy characterized by neonatal sideroblastic pancytopenia and, less frequently, severe exocrine pancreatic insufficiency with malabsorption [52]. Interestingly, infants surviving into childhood or adolescence heal from the haematological failure, but invariably accumulate deleted mtDNA species in muscle, heart and brain, thus developing the clinical features of KSS, including RP. The majority of single large-scale rearrangements of mtDNA are sporadic and are therefore believed to be the result of the clonal amplification of a single mutational event, occurring in the maternal oocyte or early during the development of the embryo [48]. It is not yet understood why, in multisystem disorders such as KSS, in which ∆-mtDNAs are virtually ubiquitous, mutations are not transmitted through female gametes to the progeny [53]. One possibility is that the germinal cells containing deleted genomes are not viable for gametogenesis and/or fertilization. However, mother-to-offspring transmission has occasionally been documented in KSS/CPEO. Therefore, a prudential figure of 5% recurrence risk is suggested in the genetic counselling of ∆-mtDNA affected women [53]. 

The relative amount and tissue distribution of the molecular lesion dictate the onset and severity of the disease. Transmitochondrial cytoplasmic hybrids (cybrids), obtained by introducing deleted mtDNAs into mtDNA-less rho^0^ cells, showed impaired respiration [54]. A threshold of >60% rearranged mtDNA molecules is enough for OXPHOS failure to occur. The more widespread is the tissue distribution of the lesion, the more severe is the clinical syndrome, from CPEO, to KSS, to Pearson’s syndrome. This notion is also relevant for the diagnosis: for instance, deletions are confined to the muscle biopsy in PEO, but in KSS they can also be found in blood, albeit in lesser amounts, while in Pearson’s syndrome the amount is comparable in blood and muscle. Most rearrangements occur across direct repeats of variable length [55], suggesting a mechanism based on illegitimate homologous recombination. Defective OXPHOS of mitochondria containing ∆-mtDNA is due to the loss of both mit (encoding mRNA translated into proteins) and syn (encoding RNA components of in situ protein synthesis) genes contained within the deletion. In particular, because the lack of tRNA genes results in incompetency for translation mitochondria containing only ∆-mtDNA cannot synthesize functional OXPHOS enzymes. However, partial correction of this impairment can be accomplished through complementation by mRNAs and tRNAs synthesized from wild-type mtDNA, provided that ∆-mtDNA and wild-type mtDNA co-segregate in the same organelles [56]. The retinal phenotype is often key to suspecting a mitochondrial disease, and specifically KSS and NARP/MILS. Cybrids have turned to be of fundamental relevance to validate the pathogenicity of mtDNA point mutations as well [57].

## 12. Genetic Heterogeneity in Mitochondrial RP

RP changes in mitochondrial disease can show considerable variability, but specific recurrent patterns allowed grouping into the following different types. In a recent survey of the genetic basis of 23 cases of mitochondrial retinopathy several interesting features have been reported [58].

Genetic testing identified sporadic large-scale mitochondrial DNA deletions or variants in mtDNA MT-TL1, MT-ATP6, MT-TK, MT-RNR1, and a deleterious homozygous mutation in the nucleus-encoded *RRM2B* gene in one subject. *RRM2B* is the ribonucleotide reductase regulatory TP53 inducible subunit M2B. 

The *RRM2B* gene encodes for the p53 inducible small subunit (p53R2), of a protein called ribonucleotide reductase (RNR). Two copies of the p53R2 subunit are attached to two copies of another protein called R1 to form RNR. (R1 can also attach to another small subunit, called R2, to make another form of RNR). Whether made with p53R2 or R2, RNR helps produce DNA building blocks (nucleotides), which are joined to one when DNA is synthesized.

Based on retinal imaging, three phenotypes could be differentiated: type 1, with mild, focal pigmentary abnormalities; type 2, characterized by multifocal white-yellowish subretinal deposits and pigment changes limited to the posterior pole; and type 3, with widespread granular pigment alterations. Advanced type 2 and 3 retinopathy presented with chorioretinal atrophy that typically started in the peripapillary and paracentral areas with foveal sparing.

No optic atrophy was detected at fundus exam or by optical coherence tomography (OCT) measures of the peripapillary RNFL thickness. Two patients of this series exhibited a different phenotype: one revealed an occult retinopathy, and the second carried a retinopathy associated with a mutation in the *RRMB2* gene (see below) with substantial photoreceptor atrophy before loss of the retinal pigment epithelium. Patients with type 1 and mild type 2 or 3 disease demonstrated good visual acuity and no symptoms associated with the retinopathy. In contrast, patients with advanced type 2 or 3 disease often reported vision problems in dim light conditions, reduced visual acuity, or both. Short-wavelength autofluorescence usually revealed a distinct pattern, and near-infrared autofluorescence was severely reduced in type 3 disease. The retinal phenotype was key to suspecting a mitochondrial disease in 11 patients, whereas 12 patients were diagnosed before retinal examination.

Nine subjects had a single, sporadic, large-scale heteroplasmic deletion of mtDNA. Seven patients carried the heteroplasmic m.3243A > G “MELAS” mutation in MT-TL1. Private mutations of mtDNA were present as m.3255G > A in MT-TL1, m.3244G > A in MT-TL1, m.1021T > C in MT-RNR1, m.9171A > G in MT-ATP6, m.8344A > G MERRF in MT-TK (MERRF stands for Myoclonus Epilepsy with Ragged-Red Fibres), and one case of multiple mtDNA deletions associated with a p.Ala192Val change in the protein encoded by the nuclear *RRMB2* gene. Surprisingly, only one NARP/MILS m.8993T > G mutation was present in this series [58]. In our own experience RP is invariant in this mutation, which is relatively frequent in mitochondrial disease patients.

## 13. Age-Related Macular Degeneration (AMD)

AMD is the leading cause of blindness among people over 60 years of age [59]. AMD is characterized first by the formation of drusen (deposits forming between the RPE and Bruch’s membrane), and it then progresses from the dysfunction and death of RPE cells to photoreceptor loss and severe visual impairment and blindness. As mentioned above, the RPE plays many roles in visual function: absorption of stray light with pigment granules, formation of the blood–retina barrier with tight junctions, transport of nutrients and ions, secretion of growth factors and transport molecules, isomerization of retinol in the visual cycle, and phagocytosis of rod outer segments (ROS) [16]. Even though AMD is a multifactorial disease affecting different retinal cell types, recent years have seen the emerging theory that mitochondrial damage to RPE due to oxidative stress may be a mechanism for AMD pathogenesis [60,61,62]. Early studies treating RPE cells with H_2_O_2_ to induce oxidative stress showed preferential damage of mitochondrial DNA and subsequently provided a rationale for a model of AMD that is mitochondrion-based [63]. There is increasing evidence of an age-related decline in mitochondrial function in people with AMD. Mitochondrial numbers, area and density of mitochondrial matrix were shown to decrease with age along with partial-to-complete loss of mitochondrial cristae [64]. Significant mtDNA damage to has been reported in subjects with AMD compared with age-matched controls [65]. Furthermore, macula-specific mtDNA damage has been shown to increase with age [66]. In RPE cells, one proposed driving cause for mtDNA damage may be the oxidative stress under conditions of imbalance in retinal homeostasis [67,68,69]. However, ROS overproduction can also be a consequence of mtDNA damage [70].

## 14. Mitochondrial Optic Atrophy

The other retinal component exquisitely vulnerable to mitochondrial dysfunction is the RGC with its long axonal projection, travelling to long distance from the cell body, all the way down to LGN. RGCs degeneration implies axonal loss and consequent optic atrophy, which is an extremely frequent feature in neurodegenerative disease, having primary mitochondrial disorders as paradigm [71]. Table 1 displays the main conditions associated with non-syndromic and syndromic optic atrophy.

The first distinction is between those diseases where optic atrophy is the only pathological feature, and syndromic phenotypes. This includes two major entities: maternally inherited Leber hereditary optic neuropathy (LHON), due to mtDNA point mutations affecting cI subunit genes, and autosomal dominant optic atrophy (DOA), which is prevalent due to heterozygous mutation in the nuclear gene *OPA1*, however also other genetic causes are also described for DOA, almost invariably related to mitochondrial function, in particular to the dynamic homeostatic properties that characterize mitochondria undergoing fusion and fission.

The syndromic forms of optic atrophy include a vast catalogue of neurodegenerative diseases, even if a continuum occurs between LHON and DOA with the so-called “plus” forms, which are multisystemic versions of the same pathogenic mechanism underlying the non-syndromic entities.

## 15. Non-Syndromic Optic Atrophy: Leber Hereditary Optic Neuropathy (LHON)

LHON remains a paradigm for mitochondrial diseases, as it was the first to be associated with a missense mutation of mtDNA in 1988 [72], being the ideal candidate due to the clear-cut maternal inheritance [73]. Currently, three key primary mutations affecting cI subunits, i.e., m.3,460G > A/*MT*-ND1, m.11,778G > A/*MT-ND4* and m.14,484T > C/*MT-ND6* are found in over 90% of patients worldwide. However, there is also a growing list of rare mtDNA mutations [74], many confirmed to be pathogenic by appropriate functional studies, others remaining in the putative status as present in single cases or lacking solid evidence for their functional impact (accessed 22/12/2021). More recently, a new genetic landscape has emerged for LHON as clinical phenocopies indistinguishable from the mtDNA-related disease are being identified as recessive traits associated with nuclear-encoded components of cI or incompletely penetrant mildly pathogenic variants in OXPHOS-related gene products, such as the Tyr51Cys of DNAJC30 gene [75,76].

As shown in Figure 6, LHON is a bilateral optic neuropathy that affects suddenly with sub-acute progression predominantly young-adult males. It was mentioned in the 19th century by Von Graefe first, but then subsequently named after Theodor Leber for his systematic description [77]. LHON prevalence, as calculated in North-East of England where referrals are centralized to a single diagnostic centre, has been estimated at 1/27,000 [78], whereas a recent meta-analysis of available studies concluded for 1/40,000 in Europe [79]. In either case, most probably LHON remains largely underestimated, as diagnosis is still frequently delayed or even not reached.

The clinical features have been recently categorized into different stages [80]. The preclinical stage, which may last lifelong without conversion to active disease, refers to unaffected mutation carriers who display some distinguishing fundus abnormalities [81], nowadays better defined at neuro-ophthalmological examination [82] and consisting of different degrees of swelling of the RNFL, as well as microangiopathic features with vessel tortuosity and artero-venous shunting [83]. A subset of these individuals, may convert to the active disease, entering the subacute stage, when patients become symptomatic complaining central vision loss due to progressively enlarging central scotoma at visual field, which impairs visual acuity. At funduscopic examination the features described in the preclinical stage are enhanced [84], with RNFL swelling and subsequent atrophy following a precise pattern of natural history, as measured by OCT [85,86], and the appearance of temporal pallor at the optic disc (Figure 7, left panel). The microangiopathy, very evident in the first weeks of evolution, tends to vanish as atrophy of the optic nerve becomes established. This stage may last a few weeks up to six months and is followed by the so-called dynamic stage, generally lasting from six months to one year from onset, when the chronic stage represents the final outcome of the active disease (Figure 7, central panel). The dynamic stage is defined by the established nadir of visual deterioration, but yet the RNFL may be swollen and apparently evolving towards normalization of thickness, which invariably leads to profound loss of axons and optic atrophy as end stage. The disease in most cases is bilateral or asymmetric with one eye losing vision first and the other following within days, weeks or months. A variable subset of patients may experience spontaneous amelioration of visual function [87], with shrinking of the scotoma and/or fenestration leading to some recovery of visual acuity. Favourable prognostic factors for visual recovery are the mutation type, the m.14,484T > C/*MT-ND6* having the best prognosis, and age at onset, as childhood onset has better outcome with all mutations [88]. Disease onset typically is at young-adult age, slightly later for females, but a childhood version of the disease is now well-described [89], as well as late and very late cases [90,91]. 

Along the maternal line only some individuals become affected, defining LHON as a disease with incomplete penetrance, and male prevalence is well established. Both these features are still under investigation, as these are not easily explained by the primary mtDNA mutation only, which represents the necessary but not sufficient genetic predisposition [92].

One first modifying factor, which has been recognized long ago, is the influence of the mtDNA background, given the wide sequence variability dependent on the population-specific mtDNA haplogroup. In the LHON patients of European descent an increased penetrance is associated with the haplogroup J [93], with some differences dependent on specific clades defined by polymorphic variants affecting cI and cIII [94,95]. 

Likewise, nuclear DNA modifying variants are assumed to play a role for incomplete penetrance, but with the exception of a few families in which the impact of genes involved in OXPHOS efficiency has been documented [96,97], there is no common universally impacting variant exerting the modifying effect in the large majority of patients identified. Also, the long-lasting hypothesis that genetic variation in the chromosome X might have explained the male prevalence, remains inconclusive [98,99]. 

Lastly, the impact of environmental factors that can trigger the disease has been now conclusively established, with tobacco smoking being the most relevant factor, followed by alcohol consumption, as well as exposure to certain toxins and drugs. As tobacco seems to exert a clear-cut effect as an environmental trigger [100,101], it has also been proposed to define the young-adult cases with a prevalent genetic impact on disease determination and the late-onset as those where tobacco smoking or other prolonged environmental exposures play a major role in precipitating disease conversion. This implies the assumption that these individuals might have remained at the preclinical stage if not exposed to the environmental triggering factor.

On the pathogenic ground, a series of in vitro studies using patient-derived fibroblasts and cybrid cells documented that cI impairment due to LHON mtDNA mutations leads to reduced bioenergetics efficiency [102], increased ROS production [103], which ultimately translates into increased propensity of cells to undergo apoptotic cell death [104,105]. The defective mitochondrial function in some LHON mutation carriers may be efficiently counteracted by the universal compensatory mechanism that cells orchestrate in mitochondrial disease, which is the increase in mitochondrial biogenesis. This was experimentally shown in vitro to be promoted by oestrogens, providing a potential explanation for the male prevalence [106]. Extending this observation to the general population of individuals carrying LHON mutations and using the mtDNA copy number as a readout of mitochondrial biogenesis, unaffected mutation carriers were clearly characterized by having the most efficient mitochondrial biogenesis when compared with the affected maternal relatives and also to a control population [107]. This establishes a potential marker of predictive value for whom is at highest risk of clinical conversion. Remarkably, tobacco smoke depresses this compensatory mechanism, reducing mtDNA copy number, thus coherently acting as disease trigger and modulator of penetrance [108].

## 16. Non-Syndromic Optic Atrophy: Autosomal Dominant Optic Atrophy (ADOA)

Unsurprisingly, given the close similarities with LHON, also ADOA was also discovered to be due to mitochondrial dysfunction, as in year 2000 two studies documented the association of most ADOA cases with heterozygous mutations in the *OPA1* gene, known for its role in mitochondrial dynamics, in particular fusion of the mitochondrial inner membrane [109,110]. In more recent times, further genes have been associated as relatively frequent cause of ADOA, including proteases indirectly implicated in OPA1 proteolytic processing such as *AFG3L2* and *SPG7* [111,112], or *ACO2*, encoding a matrix Krebs’ cycle enzyme [113] and other rarer forms linked to dominant mutations in the *OPA3* [114], *WFS1* [115], *DNM1L* [116], and *SSBP1* [117,118,119] genes. 

OPA1 is a protein of the inner mitochondrial membrane with multitasking roles. It promotes fusion of the inner mitochondrial compartment, but it is also involved in structuring the mitochondrial cristae and maintaining the physiological arrangement of the mitochondrial respiratory chain complexes. In OPA1 mutant cells cultured in galactose, a carbon source that is bioenergetically exploited only through oxidative phosphorylation, mitochondria become fragmented, as shown in Figure 8A. The cristae are disrupted or sparse by TEM analysis (Figure 8B).

As clinical entity, the definition of dominant optic atrophy with infantile onset, beyond previous scattered clinical reports, was established by the Danish ophthalmologist Paul Kjer [121]. Most of these Danish families were later shown to be due to *OPA1* mutations, with a relevant founder effect. DOA prevalence, again based on epidemiological survey in the North-East of England, is set at 1/25,000 [122] suggesting that DOA equals or might even be more frequent optic atrophy than LHON.

Clinically, ADOA differs from LHON for an insidious onset almost invariably before age ten, with very variable natural history, which frequently is characterized by a substantial stability over decades, or relentless progression, sometimes stepwise [29,30,31]. As shown in Figure 7 (right panel), funduscopic features, as well as OCT studies describe pale optic disc with significant reduction in RNFL thickness, marked on the temporal sector remarking, as in LHON, a predilection for the papillomacular bundle. Visual fields show variable degrees of central scotoma and visual acuity reduction may be extremely variable, from asymptomatic cases denoting incomplete penetrance, to very severe visual impairment. The better-established genotype-phenotype relationship concerning this variable clinical expressivity is that mutations leading to haploinsufficiency are usually milder than missense mutations, in particular if the latter affect the GTPase domain [30,31]. 

## 17. Syndromic Optic Atrophy

### “Plus” Forms of LHON and ADOA

Both LHON and ADOA may be associated with multisystemic disease, which are frequently referred as LHON plus [123] or ADOA plus [124].

LHON, either with the three primary common mtDNA mutations, but more frequently with rarer mtDNA mutations again affecting the mtDNA-encoded cI subunit genes, may present with a spectrum of clinical manifestations ranging from optic atrophy plus dystonia and basal ganglia bilateral lesions resembling Leigh syndrome to MELAS syndrome [29,30]. Other reported associations may include cerebellar atrophy [125], myoclonus [126], myelopathy [127,128] and peripheral neuropathy [129]. Perhaps the most puzzling and controversial association is the co-occurrence of LHON with multiple sclerosis, for which it is still debated if this must be considered merely a coincidence based on prevalence of both disorders or if one may be triggering the other [130,131]. One disorder is likely to impact the natural history of the other [132], and this entity, also known as “Harding disease” [133], needs further investigation.

ADOA has been known to be associated with sensorineural deafness as occasional co-occurring feature [134], but in 2008 an unexpected observation was reported, that missense heterozygous OPA1 mutations affecting the GTPase domain could lead to a multisystem disorder characterized by chronic progressive external ophthalmolplegia (CPEO), peripheral neuropathy, sensorineural deafness, cerebellar atrophy, white matter lesions and myopathy with RRF due to accumulation of mtDNA multiple deletions, providing a mechanistic link of OPA1 with mtDNA maintenance disorders [135,136]. This phenotype may also have optic atrophy as an underscored feature, whereas age-related neurodegeneration leading to parkinsonism and dementia may be prevalent [137]. These multisystem forms, frequently hallmarked by CPEO and mtDNA maintenance disorder, were further associated with other genes also involved, directly or indirectly, in mitochondrial dynamics, such as MFN2 [138], AFG3L2 [139], and SPG7 [140]. Interestingly, within the range of DOA plus, similarly to LHON, there again emerged an association with multiple sclerosis, reissuing the same considerations as for LHON [141,142]. 

In addition to DOA, OPA1 has been implicated also in biallelic froms of syndromic optic atrophy. Behr syndrome, for example, first described in 1909 by the ophthalmologist Carl Behr, designates an autosomal recessive condition characterized by early-onset optic atrophy along with neurological features, including ataxia, spasticity, and intellectual disability [143]. Other signs and symptoms may be present and vary from case to case. This condition is caused by mutations in the OPA1 gene [144], but also by mutations in the C19ORF12 gene, encoding a protein responsible of a subtype of NBIA, mitochondrial membrane protein-associated neurodegeneration (MPAN), which is synonymous of Behr’ syndrome [145].

## 18. Rare Mitochondrial Causes of Optic Atrophy

### 18.1. Optic Atrophy and Charcot–Marie–Tooth (CMT) Peripheral Neuropathy

The OPA1 fellow proteins implicated in the mitochondrial fusion machinery are mitofusins 1 and 2 encoded by *MFN1* and *MFN2* genes [146]. While OPA1 is deputed to fusion of the inner mitochondrial membrane, MFN1 and 2 are considered key to fusion of the outer mitochondrial membrane, where both are located. However, MFN2 has also been reported to associate with the endoplasmic reticulum (ER) [147], implicating a possible role of MFN2 in the ER–mitochondrial interaction, through the mitochondrial-associated membranes (MAMs) [148]. *MFN2* mutations were originally described as causative of Charcot–Marie–Tooth neuropathy type 2A in both dominant (CMT2A2A) and recessive form (CMT2A2B) [149,150]. In many cases optic atrophy is a common feature and this association has also been named as hereditary motor and sensory neuropathy VI (HMSN6/CMT6) [151,152]. As mentioned in the previous section MFN2 dominant mutation may also associate with the ADOA *plus* phenotype [138], supporting a continuum of phenotypes as a converging pathogenic mechanism may come from defects of different components of the same functional machinery, mitochondrial dynamics in this case.

Another outer mitochondrial membrane protein, the degenerated carrier SLC25A46, which has been proposed to be the human homologue of yeast protein Ugo1, is implicated in mitochondrial fission and membrane lipids homeostasis [153,154]. Dominant mutations in the *SLC25A46* gene have been initially associated with a spectrum of optic atrophy phenotypes, usually in combination with axonal CMT (HMSN6B/CMT6B) [153], subsequently expanding the spectrum of clinical phenotypes to severe cases of infantile Leigh syndrome, HMSN6 [155] and forms of pontocerebellar hypoplasia (PCH1E) [156].

### 18.2. Syndromic Optic Atrophy Implicating cI and Altered Interorganellar Cross-Talk—TMEM126 and RTN4P1 Genes

As cI dysfunction is central to LHON, and prominent cI impairment is also detected in OPA1-related ADOA, it is not surprising that mutations in other cI-related genes may lead to optic atrophy when mutated. *TMEM126A* is one of them, recently identified as an assembly factor for ND4 subunit [157,158]. Mutations in *TMEM126A* cause recessive optic atrophy (OPA7), isolated or with sensorineural hearing loss, mostly found in families of Maghrebian ancestry due to a founder event [159,160]. 

Another protein, which is formally involved in mitochondria-ER tethering with a poorly understood role, is RTN4IP1 that interacts with RTN4/NOGO present on ER membrane and enriched in MAMs [161]. Recessive mutations in *RN4IP1* gene cause isolated (OPA10) or syndromic optic atrophy, the latter associated with ataxia, mental retardation, and seizures, and severe encephalopathy at the extreme end of the spectrum [162]. Recent reports further highlighted the adjunctive occurrence of rod-cone dystrophy associated with optic atrophy in *RN4IP1* patients [163,164]. Remarkably, patients’ fibroblasts show mitochondrial network fragmentation and respiratory cI disassembly, bridging the known optic atrophy-related pathogenic mechanisms and prompting further investigations to disentangle how MAM, cI and mitochondrial dynamics converge physiologically and lead to disease when impaired [165].

## 19. Partial Depletion of mtDNA Introduces a New Paradigm—The Cases of SSBP1 and LIG3 Genes

Until recently, the mitochondrial depletion syndromes were considered as almost invariably associated with infantile severe phenotypes, either encephalo-myopathies, but also with organ specificity, as liver and kidney may be selectively targeted [166]. 

This paradigm has been changed by the identification of mostly dominant, but also recessive, mutations affecting the SSBP1 protein, encoding the mitochondrial single-stranded DNA-binding protein that is essential for mtDNA replication, associated with optic atrophy (OPA13), retinal dystrophy with foveopathy, but also with deafness, myopathy and a remarkable nephropathy leading to kidney failure requiring transplantation [117,118,119]. Partial mtDNA depletion and defective mtDNA maintenance has been well documented in these patients, adding to the infantile forms of mtDNA depletion syndromes also an adult disease.

Likewise, recessive mutations in another key factor for mtDNA replication, the ligase LIG3, have also been associated with a syndrome including retinal dystrophy and severe encephalomyopathy with gastrointestinal dysmotility and stroke-like episodes reminiscent of MELAS [118,119]. Again, partial mtDNA depletion in post-mitotic tissues is the leading feature in these patients, which may become affected early in life but survive to adult age.

## 20. Therapeutic Strategies

In this review we shall consider only specific therapeutic approaches, rather than the usual antioxidant and vitamin cocktails usually recommended for mitochondrial diseases, whose efficacy is generally very limited.

Most of the therapeutic attempts have focused on LHON, although several years ago it was proposed, and proven in cells to be effective, the use of a recombinant, mitochondrion-targeted restriction enzyme SmaI, which recognizes the GGGCCC motif formed by the m.8993T>G mutation. Once internal to mitochondria, the restriction enzyme destroys mutant mtDNA molecules and promoting the proliferation of wt-mtDNA in NARP/MILS cells shifts the original mutant heteroplasmy under the threshold for OXPHOS dysfunction [167]. Currently, the preclinical development on the shifting of heteroplasmic mtDNA mutations, including those implicated in NARP and KSS, is ongoing using zinc-finger nucleases (ZFNs), TALEN-based nucleases (TALENs), and more recently meganucleases, all engineered to target mitochondria by an N-terminal mitochondrial targeting sequence (MTS) [168,169]. As these gene therapy strategies are approaching the stage of clinical trials in humans, it is likely that the retina will be one of the first tissues to be targeted.

Currently, idebenone, a short-chain benzoquinone, is the only disease-specific drug approved to treat visual impairment in adolescents and adults with LHON.

The mechanism of action of idebenone involves its antioxidant properties and ability to act as a mitochondrial electron carrier. Idebenone overcomes mitochondrial complex I respiratory chain deficiency in patients with LHON by transferring electrons directly to mitochondrial complex III (by-passing complex I), thereby restoring cellular energy (ATP) production and re-activating inactive-but-viable RGCs, which ultimately prevents further vision loss and promotes vision recovery [170]. The approval of idebenone in the treatment of LHON was based on the overall data from a randomized clinical trial [171], retrospective assessment of off-label treated patients [172] and real-world data obtained from an expanded-access program [173]. Taken together, these studies provide convincing evidence that oral idebenone 900 mg/day for 24 weeks has persistent beneficial effects in preventing further vision impairment and promoting vision recovery in about 50% of patients with LHON relative to the natural course of the disease. Therefore, idebenone is the gold standard to treat visual impairment in adolescents and adults with LHON [174,175,176].

A definitive cure for this and other mitochondrial diseases will probably come from gene therapy. The results obtained on other diseases, such as spinal-muscle atrophy [177] are very encouraging As compared to nuclear DNA, mitochondrial DNA gene therapy poses additional problems, including the fact that (i) mitochondria are surrounded by a double membrane, (ii) with the exception of non-syndromic LHON, primary mitochondrial diseases are often multisystemic, making it difficult to reach the high titres required to target a majority of cells in all the affected organs, (iii) mammalian mtDNA has no recombination systems, preventing the use of homologous recombination-based approaches, (iv) mitochondria cannot import nucleic acids, preventing the use of CRISPR/Cas9-based techniques, and (v) last-but-not-least, mtDNA is polyplasmic instead of the diploid or haploid organization of nuclear genes, which implies an extremely high number of genomes to be targeted in each cell to produce a tangible effect [87]. Nevertheless, the recent development of approaches to shift mtDNA heteroplasmy opened great therapeutic opportunities. However, this approach can be exploited only in case of heteroplasmic mtDNA mutations. In most LHON cases, the mutations are homoplasmic, and therefore unsuitable for this treatment. In addition, as previously mentioned, the CRISPR/Cas9 system, which has great potential for the manipulation of the nuclear genome, cannot be applied to mtDNA due the impossibility of importing either DNA or RNA (such as the guided RNAs which are integral components of the Cas9 nucleoprotein), into mammalian mitochondria [178].

A distinct advantage of non-syndromic LHON is that the RGC layer of the retina is affected, which can be approached by mildly invasive techniques, such as intra-vitreal injection for gene therapy delivery. Another advantage is that the availability of two eyes theoretically allows for designing a clinical trial by treating only one retina and compare the effects, taking as a control the contralateral, untreated retina. A third important advantage is that visual acuity and field as well as retinal thickness can be measured quantitatively, thus providing precise metrics to assess the initial conditions at the start of the treatment and reliably measure the possible variations during the follow up to certify the effectiveness of treatment.

In fact, there are currently several adeno-associated vector (AAV)-based clinical trials registered for LHON, some being completed, using the so-called allotopic expression of ND4, i.e., the expression of the wild-type version of the mtDNA-encoded protein from the nucleus. The transgene contains, in addition to a promoter, one or multiple MTS, allowing for the proteins translated by cytoplasmic ribosome to be translocated with the mitochondria to complement the endogenous mutant ND4. This strategy was originally proven in cybrids for both the NARP m.8993T>G mutation [179] and the common LHON m.11778G>A mutation [180]. Although preclinical evidence that allotopically expressed mtDNA-encoded subunits can be efficiently assembled into respiratory complexes has been debated, [181,182], two Phase 3 double-masked, randomized, sham-controlled trials with an AAV-2 serotype vector are now concluded, providing some encouraging results [183,184], as compared with natural history of LHON, but also highlighting an unexpected and puzzling event. Despite the recovery of visual function, which was previously thought unlikely for LHON according to natural history [185], is that this occurred bilaterally in both eyes, and even if in favour of the treated eye this leads to the failure of the primary endpoint, which was a prespecified consideration for significance of the inter-eye comparison of treated and untreated eyes [185]. This has been thought to be due to the leakage of the AAV vector into the contralateral eye, as shown experimentally in non-human primates [186]. Consistently, another two groups running similar trials with parallel gene therapy products have reported similar results [187,188]. In addition to this bilateral effect could be the presence of a secondary mutation in mtDNA, which could be a prompt to rethink the clinical trial design for such approaches in the future. An approval of this therapy is pending with EMA, as is the case for the results of a third Phase 3 trial with bilateral intra-vitreal injection, which have been announced only by a press release of the company sponsoring the study at this stage.

The therapeutic options for OPA1-related DOA are still in a very preliminary phase [189], with some encouraging results obtained by off-label idebenone administration in DOA patients [190]. At the preclinical stage numerous research pipelines are active, from screening molecules for repurposing [191], mechanistic studies to define therapeutic targets [192,193], and gene therapy strategies [194], including the recent gene editing of a frequent OPA1 missense mutation [195]. Clinically DOA does not have the confounder of spontaneous recovery of visual function as for LHON, but it is also a slowly progressing or substantially stable disease that questions are posed on which primary endpoint most reliably may reflects therapeutic efficacy. Finally, a complication is due to the processing of OPA1 in multiple molecular isoforms, which are present in different ratios depending on the cell type, providing a substantial complexity on the issue of the therapeutic molecular manipulations. 

For a more detailed description of therapeutic approaches for mitochondrial optic neuropathies, we refer the reader to a comprehensive review that has been recently published [195].

## 21. Concluding Remarks

As frequently occurs in mitochondrial medicine, the dichotomy between defects leading only to optic atrophy (LHON and DOA) as opposed to those leading to retinal dystrophies affecting both photoreceptors and retinal pigmented epithelium (KSS and NARP) is a paradigm that has been blurred by the most recent findings. This was originally ascribed to a prevalent defect increasing ROS production as opposed to a clear-cut bioenergetics defect. This has been clearly contradicted by the *SSBP1* gene example, where mtDNA depletion leads to defective bioenergetics, leading simultaneously to optic atrophy and retinal dystrophy, as well as the case of *RTN4IP1* mutations and other rare conditions. Likewise, the long-standing debate between *lumpers* and *splitters* may be considered now overcome by the tremendous degree of overlapping phenotypes, with converging pathogenic mechanisms including OXPHOS deficiency, ROS production, mitochondrial dynamics, inter-organellar interaction such as ER mitochondria, which are all involved in the higher-order homeostatic control of mitochondria implying their biogenesis and removal.

The eye remains the tip of the iceberg of mitochondrial medicine and the most convenient playground to understand mitochondrial-related neurodegenerative mechanisms. As such, mitochondrial eye retinal disorders, being part of the central nervous system, will possibly be the first to reach the stage of implementation of effective, easily accessible and measurable therapeutic strategies.

## Figures and Tables

**Figure 1 ijms-23-00210-f001:**
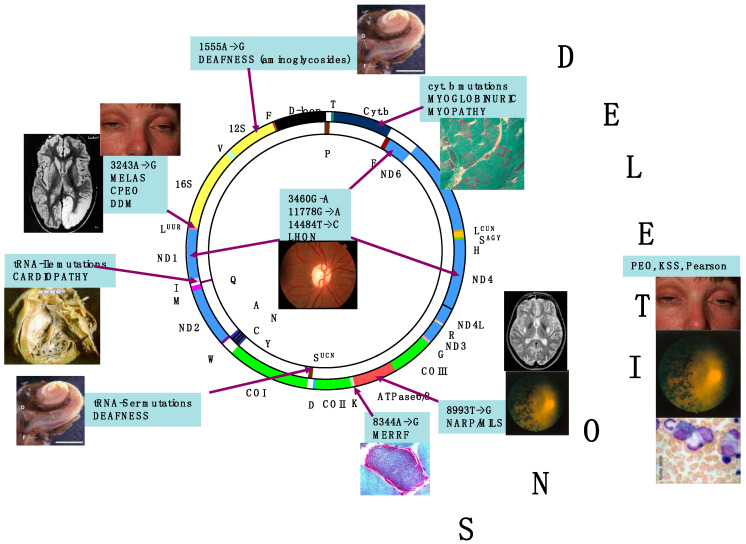
Morbidic map of mtDNA mutations. The scheme of the circular 16.5 kb human mtDNA is depicted, with several clinical and molecular syndromes associated with mutations of the molecule. Genes encoding subunits belonging to the same MRC complex have identical colours. In yellow are indicated the genes encoding the 12S and 18S ribosomal RNAs. Different colours and aminoacids expressed in the single-letter code designate the tRNA-encoding genes. The D-loop control region is in black. LHON is associated predominantly with three-point mutations, usually homoplasmic, in different genes of cI, but other, rare mutations have also been reported. Deletions, associated with KSS and other syndromes including adult-onset progressive external ophthalmoplegia and neonatal Pearson’ syndrome, usually affects the “major” arc occupying approximately two thirds of the mtDNA circle, included between the D-loop and the WANCY cluster where the origin of light strand replication is contained.

**Figure 2 ijms-23-00210-f002:**
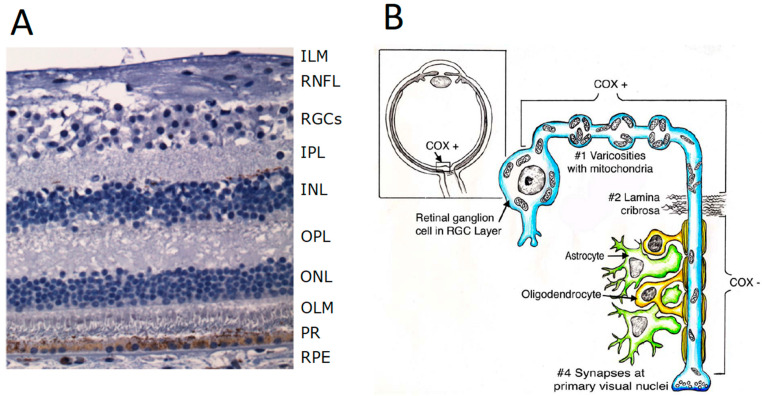
Anatomy of the retina and its connections. (**A**) A section of normal retina. ILM: inner limiting membrane; RNFL: retinal nerve fibres layer; RGCs: retinal ganglion cells; IPL: inner plexiform layer; INKL: inner neuronal layer; OPL: outer plexiform layer; ONL: outer neuronal layer; OLM: outer limiting membrane; PR: photoreceptors (inner segments); RPE: retinal pigmentary epithelium (modified from Maresca A, Carelli V, 2021 [18]). (**B**) The structure of the eyeball is outlined in the upper left inset and a retinal ganglion cell is outlined with the part of the axon (proximal to the lamina cribrosa, unmyelinated), as well as the initial part of the axon after the emersion from the lamina cribrosa, where it is myelinated by oligodendrocytes of the optic nerve. Notice the high number of mitochondria in the cell body and unmyelinated portion of the axon, often organized in mitochondrial “varicosities” compared with the few mitochondria present in the Ranvier nodes in the myelinated part of the axon (modified from Carelli V et al., 2004 [19]).

**Figure 3 ijms-23-00210-f003:**
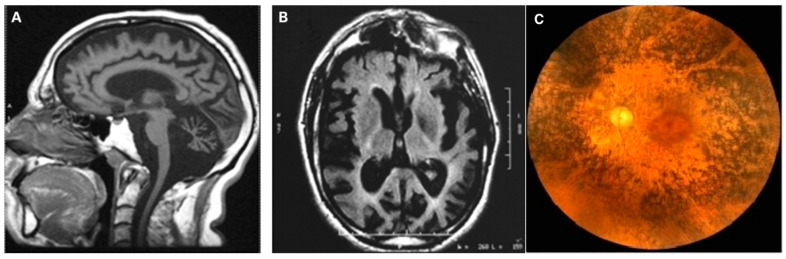
NARP/MILS. (**A**,**B**). FLAIR T2-weighted sagittal and transverse sequences of a patient with NARP/MILS (modified from Lopez-Gallardo, 2009 [46]). Profound cortical atrophy with enlargement of the ventricular system; severe cerebellar atrophy. (**C**). Fundus oculi showing retinal dystrophy in a patient with NARP/MILS.

**Figure 4 ijms-23-00210-f004:**
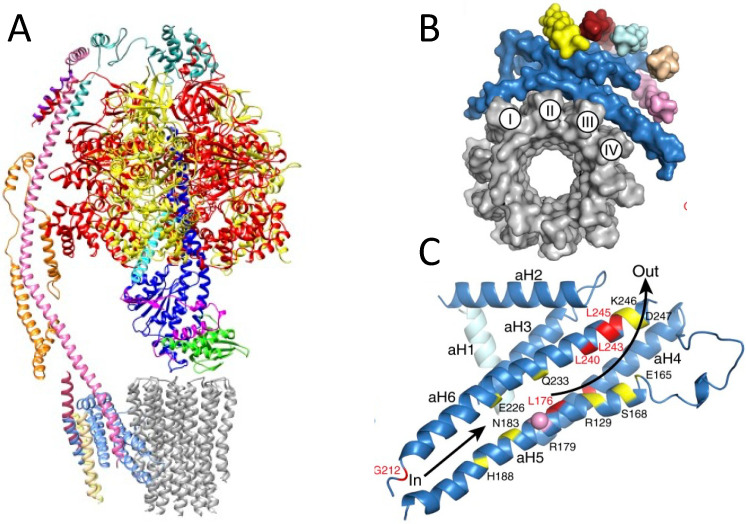
Molecular features of ATP synthase from *Pichia angusta* (modified from Vinothkumar et al. [47]). (**A**) Structure of the F-ATPase from *P. angusta*. The α-, β-, γ-, δ-, and ε-subunits forming the membrane extrinsic catalytic domain are red, yellow, royal blue, green, and magenta, respectively; the inhibitor protein is cyan; and the peripheral stalk subunits OSCP, b, d, and h are sea-green, pink, orange, and purple, respectively. In the membrane domain, the c_10_-rotor is grey, the resolved region of the associated subunit a is corn-flower blue. Chains Ch1–Ch4 are pale yellow, brick-red, pale cyan, and beige, respectively, and have been assigned as transmembrane α-helices in subunit f and, in ATP8, as aH1 and bH1, respectively. (**B**) The F-ATPase from *P. angusta*. The a-subunit (encoded by ATPase6) is corn-flower blue. The c_10_-ring is grey, the b-subunit (upper part not shown) is pink, and the pale yellow, brick-red, light cyan, and beige segments are transmembrane α-helices, Ch1–Ch4 assigned to subunit f, ATP8, aH1, and bH1, respectively. In the c-ring, I–IV indicate the four transmembrane C-terminal α-helices in contact with subunit a. (**C**) The a-subunit viewed from outside and looking out from the interface with the c-ring, respectively, with aH1 in pale cyan. Conserved polar residues are yellow; the positions of human mutations associated with pathologies are red. L176 in the cartoon correspond to the L156 (Leu156) mutated in the NARP mutation (either into an Arg or into a Pro). The pink sphere denotes the conserved Arg179 in aH5 that is essential for proton translocation and corresponds to human R159 residue (see text). The lower arrow indicates the inlet pathway for protons that transfer to Glu59 in the C-terminal α–helix-II of the c-ring. They are carried around the ring by anticlockwise rotation, as viewed from above, until they arrive at Arg179 where they enter the exit pathway (upper arrow).

**Figure 5 ijms-23-00210-f005:**
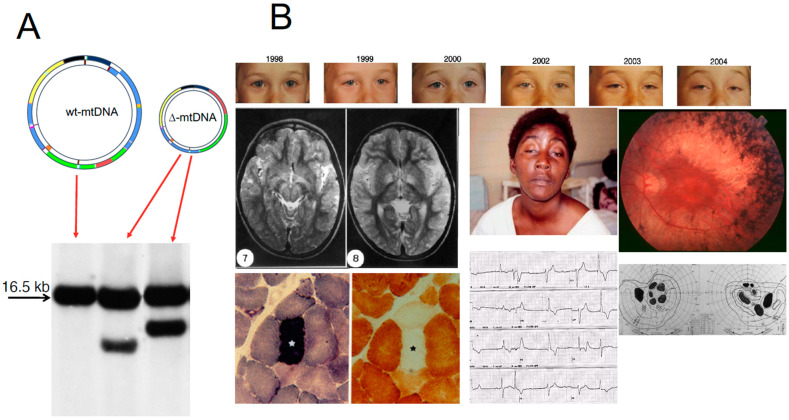
KSS: molecular and clinical synopsis. (**A**). Cartoon displaying the structure of wild-type and deleted mtDNA molecules, linearized, and subjected to Southern-blotting with audioradiography against purified radiolabeled human mtDNA. The upper band present also in the wild-type mtDNA indicates linearized normal mtDNA (wt-mtDNA). The lower bands present in the other two lanes corresponding to muscle DNA from KSS patients are deleted species (Δ-mtDNA) co-existing with wt mtDNA (heteroplasmy). (**B**). In the upper set of panels progression of bilateral ptosis in a KSS adolescent. Two FLAIR-T2 transverse brain sequences display multiple lesions. A transverse section of a muscle biopsy shows a ragged-blue fibre (SDH-hyperintense) associated with several COX-negative fibres. The lady depicted in the figure displays bilateral eyelid ptosis (with ophthalmoparesis) and facial weakness. Her ECG shows numerous extrasystolic aberrant impulses. On the left, the fundus oculi shows retinal dystrophy, with irregular defects of both visual fields (bottom).

**Figure 6 ijms-23-00210-f006:**
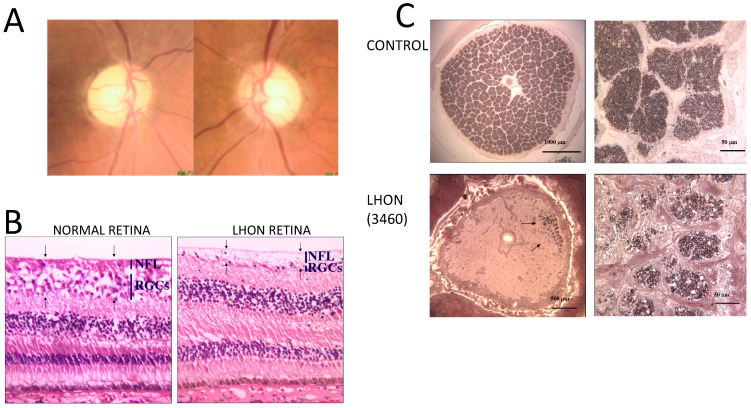
The retina in LHON. (**A**) Bilateral optic atrophy in chronic LHON. (**B**) In LHON the retinal neuronal fibres layer (NFL) and retinal ganglion cells (RGCs) layer are severely atrophic. Modified from from Maresca A and Carelli V, 2021 [18], H&E staining. (**C**) Loss of nerve fibres in a transverse section of the optic nerve of an LHON patient with the MT-3460 mutation, and in a magnified detail, compared with a control; Courtesy of Alfredo A. Sadun and Fred N. Ross-Cisneros.

**Figure 7 ijms-23-00210-f007:**
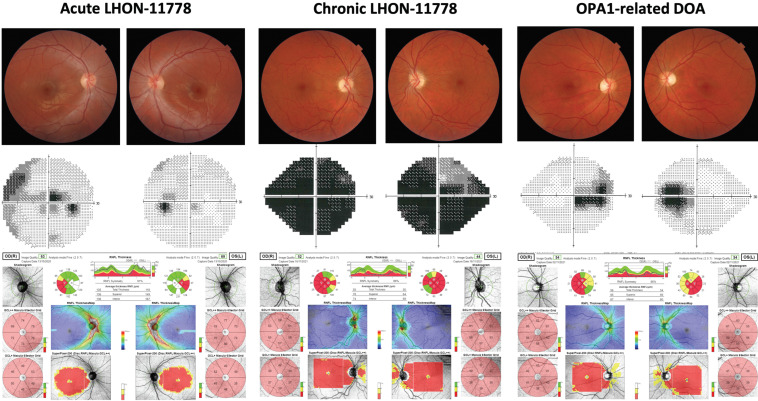
Instrumental analysis of optic atrophy. Left panel: Acute LHON-11778. Fundus pictures (upper row) of both eyes reveal an initial loss of fibres in the temporal sector corresponding to the papillomacular bundle, whereas superior, inferior and nasal quadrants present swollen fibres as denoted by the translucid appearance. The loss of temporal fibres is reflected in the visual fields (middle row) displaying enlarged blind spot and central scotoma. Finally, the RNFL thickness, as measured by OCT (lower row), clearly highlights the reduction of thickness in the temporal sector (T, red colour), whereas in the remaining sectors the thickness is at the upper limit, reflecting the swelling of axons (S, I, N, green colour). Central panel: Chronic LHON-11778. Fundus pictures (upper row) of both eyes reveal a completely pale optic disc, denoting the atrophy of the optic nerve. This is reflected by the generalized loss of sensitivity at visual fields (middle row), and reduced RNFL thickness essentially in all quadrants (T, S, N, I, red colour), with the exception of some spared axons in the super-nasal sector (S-N, green colour). Right panel: OPA1-related DOA. Fundus pictures (upper row) of both eyes reveal pale optic discs, denoting the generalized atrophy of the optic nerves. This is reflected by the dense caecocentral scotomas at visual fields (middle row), and reduced RNFL thickness essentially in all quadrants (T, S, N, I, red colour or borderline yellow-green), particularly marked on the temporal sector (T, red colour).

**Figure 8 ijms-23-00210-f008:**
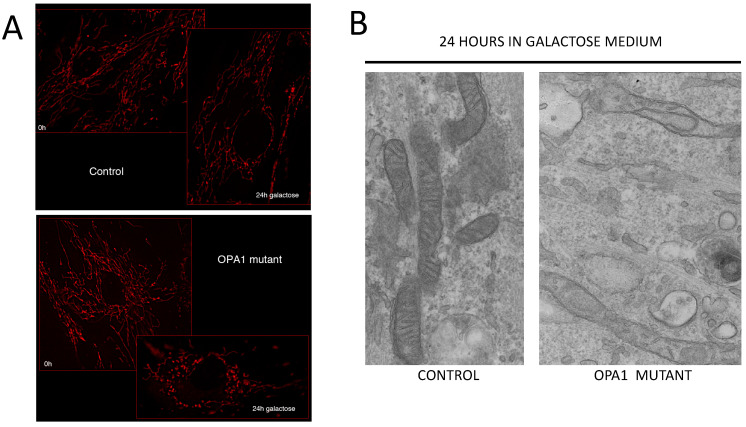
ADOA. (**A**) immunofluorescence with TMRM, a mitochondrion-specific dye. After 24 h exposure to a medium containing galactose, a carbon source that is energetically exploited only through OXPHOS, the mitochondrial network is similar to that of cells in normal glucose-containing medium in control cells, but appears severely fragmented in OPA1-mutant cells (see text for details). (**B**) The same experiment investigated the shape and internal structure of mitochondria. In OPA1 mutant cells, mitochondria are sparse and the cristae organization is severely disrupted, compared with a control sample (detail inset modified from supplemental material from Zanna et al., 2008 [120]).

**Table 1 ijms-23-00210-t001:** Genes involved in mitochondrial retinopathies.

Retinal Degeneration				
Gene mutation	Protein change	Main retinal lesions	Syndrome	OMIM
Heteroplasmic 8993T>G in mt-ATPase6	Leu156Arg in protein A of H^+^-ATP synthase	Retinal degeneration (retinitis pigmentosa)	NARP/MILS Ataxia and muscle weakness in NARP, maternally inherited Leigh syndrome if heteroplasmy is ≥70%	OMIM #551500.
Heteroplasmic 8993T>C in mt-ATPase6	Leu156Pro in protein A of H^+^-ATP synthase	Retinal degeneration (retinitis pigmentosa)	NARP/MILS Ataxia and muscle weakness in NARP, maternally inherited Leigh syndrome if heteroplasmy is ≥90%	OMIM #551500.
Heteroplasmic single, sporadic large-scale mtDNA deletions	Ablation or disruption of mtDNA genes including at least one tRNA gene	Retinal degeneration (retinitis pigmentosa)	Juvenile KSS or neonatal onset Pearson’ syndrome if children that overcome the haematological failure	OMIM #530000 OMIM #557000
**Optic Atrophy** **(non-syndromic)**				
MTND6*LDYT14459A MTND4*LHON11778A MTND1*LHON3460A MTND6*LHON14484C MTCYB*LHON15257A MTCO3*LHON9438A MTCO3*LHON9804A MTND5*LHON13730A MTND1*LHON4160C MTND2*LHON5244A MTCOI*LHON7444A MTND1*LHON3394C MTND5*LHON13708A MTCYB*LHON15812A MTND2*LHON4917G MTND1*LHON4216C MTND1*LHON4136G MTATP6*LHON9101C MTND4L*LHON10663C	A72V R340H A52T M64V D171N	Maternally inherited LHON Primary ascertained mutations: MTND5*LDYT14459A MTND4*LHON11778A, MTND1*LHON3460A, MTND6*LHON14484C MTCYB*LHON15257A Three additional putative mutations are: MTND5*LHON13730A; MTCO3*LHON9438A MTCO3*LHON9804A. Nine other mutations are ‘secondary’ mutations which may interact with the primary mutation to clinical risk. Among the more important of these mutations are: MTND5*LHON13708A; MTND1*LHON3394C; MTCO1*LHON7444A MTND1*LHON4160C MTND2*LHON5244A	Four mutations are considered ‘primary’ mutations, the presence of which greatly increases the probability of blindness. One mutation is associated with LHON plus dystonia (LDYT).	OMIM #535000
*DNAJC30* c.152A>G c.232C>T c.302T>A	DNAJC30 Y51C P78S L101Q	Autosomal recessive LHON. The protein is a chaperone involved in turnover and repair of cI.	All hallmarks of mtLHON are recapitulated, including incomplete penetrance, male predominance, and significant idebenone responsivity.	OMIM #619382
*Heterozygous OPA1 Mutations*	Truncation, aberration or single aminoacid replacements in OPA1 due to mutations in the corresponding gene	Autosomal dominant optic atrophy (ADOA), probably caused by haploinsufficiency.	Missense mutations in the GTPase domain may often determine an autosomal dominant syndrome with multiple mtDNA deletions leading to muscle weakness with or without peripheral neuropathy, bilateral eyelid ptosis, progressive external ophthalmoplegia and occasionally parkinsonism.	OMIM #165500
Recessive *OPA1* mutations, usually in compound heterozygosis	S256R + Q285R c.2708delTTAG + I382M E487K + I383M Etc.	Behr’ syndrome	Besides early-onset optic atrophy, there is ataxia, pyramidal signs, spasticity, and mental retardation	OMIM #210000
*C19ORF12*	C19ORF12	Behr’ syndrome (mitochondrial membrane protein-associated neurodegeneration, MPAN)	Besides optic atrophy in the context of Behr’ syndrome, mutations of this gene cause neurodegeneration with brain iron accumulation NBIA4	OMIM #614298
Locus Xp11.4-p11.21	unknown	OPA2	Besides very slow progression of optic atrophy, affected males may have mental retardation and minor neurological abnormalities	OMIM #311050
*OPA3*	OPA3 Possibly an outer mitochondrial membrane lipid metabolism regulator	Autosomal dominant optic atrophy	Cataract, extra-pyramidal signs, 3-methyl-glutaconic aciduria type III	OMIM #165300
Locus 18q12.2-q12.3	unknown	OPA4	Mutation found in one German family with autosomal dominant optic atrophy associated with Kidd blood group	OMIM #605293
*DRP1*	DNM1L	OPA5	Autosomal dominant encephalopathy due to defective mitochondrial and peroxisomal fission 1	OMIM #603850
Locus 8q21-q22	unknown	OPA6 Found in a consanguineous family of French origin with 4 sibs affected by early-onset, slowly progressive isolated optic atrophy. Disease progression was very slow, with moderate photophobia and dyschromatopsia.	No other clinical sign.	OMIM #258500
*TMEM126A*	TMEM126A assembly of cI	OPA7 Autosomal recessive juvenile-onset optic atrophy characterized by severe bilateral deficiency in visual acuity, optic disc pallor, and central scotoma.	Occasional sensory-motor axonal neuropathy with focal demyelinating abnormalities.	OMIM #612989
Locus 16q21-q22	unknown	OPA8	Occasionally, bilateral sensorineural hearing loss for high frequencies, with abnormal BAEP and SEP. Mitral valve prolapse or insufficiency. Subsarcolemmal accumulations of mitochondria with impaired growth of fibroblasts in galactose, and an abnormally high rate of fusion activity, suggestive of mitochondrial dysfunction.	OMIM #616648
*ACO2*	ACO2 Encodes the mitochondrial aconitase, part of the TCA cycle. In yeast ACO2 is essential for mitochondrial DNA maintenance independent of its catalytic activity.	OPA9 Recessive, early onset optic atrophy.	Onset between ages 2 and 6 months with truncal hypotonia, athetosis, seizures, in addition to the ophthalmologic abnormalities, profound psychomotor retardation, with only some achieving rolling, sitting, or recognition of family. Brain MRI showed progressive cerebral and cerebellar degeneration.	OMIM #100850
*RTN4IP1*	RTN4IP1 Reticulon, or NOGO-interacting mitochondrial protein.	OPA10 Autosomal recessive optic atrophy.	Ataxia, mental retardation, and seizures,	OMIM #610502
*YME1L1*	YME1L1 A mitochondrial metalloprotease involved in the quality control of mitochondrial proteins.	OPA11 Autosomal recessive optic atrophy in a single Saudi Arabian family.	In vitro functional expression studies and studies of patient cells showed that the mutation resulted in degradation of the mutant protein, abnormal processing of YME1L1 substrates PRELID1 and OPA1, increased mitochondrial fragmentation, and impaired cell proliferation.	OMIM #607472
*AFG3L2*	AFG3L2 A mitochondrial AAA-protease involved in quality control of mitochondrial proteins.	OPA12 Autosomal dominant optic atrophy.	Autosomal dominant spino-cerebellar ataxia (SCA28), and autosomal recessive spastic ataxia.	OMIM #604581
*SSBP1*	SSBP1 Single-stranded mtDNA binding protein, essential for mtDNA replication.	OPA13 Autosomal dominant optic atrophy with retinal and foveal degeneration.		OMIM #600439
*SPG7*	Paraplegin An AAA mitochondrial metalloprotease involved in the quality control of mitochondrial proteins.	Autosomal recessive optic atrophy	Most commonly associated with autosomal recessive HSP7	OMIM #607259
*MFN2*	MFN2	Dominant optic atrophy	CMT2A	OMIM #609260
*SLC25A46*	SLC25A46 (ortholog of yeast UGO1)	Dominant optic atrophy	CMT6B, Leigh syndrome, pontocerebellar hypoplasia (PCH1E)	OMIM #601152
*LIGIII*	LIG III An essential mitochondrial ligase involved in termination of mtDNA replication, it is present also in the nucleus.	Recessive optic atrophy.	Gut dysmotility and MNGIE-like neurological abnormalities including leukoencephalopathy, epilepsy, migraine, stroke-like episodes, and neurogenic bladder. In one family recessive mutations in LIG3 caused neonatal fatal myopathy with profound mtDNA depletion.	OMIM *600940.
Miscellaneous mutations in mtDNA and mitochondrion-related nuclear genes	Subunits of the MRC, as well as accessory proteins involved in the formation, turnover and quality control of MRCcomplexes and OXPHOS. The most frequent protein involved is SURF1, a putative chaperone of cIV.	Non-syndromic optic atrophy.	Necrotizing Encephalomyelopathy or Leigh Syndrome	OMIM #256000

## Data Availability

The data mentioned in this review are reported in the publications listed in the References chapter.

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
