# Peer review of "Mitochondrial Retinopathies"

_ijms, 2021, doi:10.3390/ijms23010210_

Round 1

Reviewer 1 Report

The manuscript “MITOCHONDRIAL RETINOPATHIES" by Zeviani and Carelli reviews the role of mitochondrial malfunctions in retinal pathology.  While this topic could be of potential interest for the readership of IJMS – in fact, it could be very impactful – several aspects of this manuscript prevent me from recommending publication in its current form.

  • It appears that the authors did not proof-read their work or mistakenly submitted an incomplete draft as is evident from multiple “XY” and “…” placeholders. All but Figure 4 are missing citation of the original artwork. Figures 5-8 have grossly inadequate legends. Several include “see text for details” without such details. Most figures are never cited in the text.
  • It is clear that the authors intended to cover only hereditary mitochondrial malfunctions as the manuscript does not mention other mitochondrial disorders, such as growing recognition of the role of mitochondria in diabetic retinopathy, for example, including mitochondrial epigenetics.  While the authors are free to choose the narrower topic, of course, it should be clearly delineated in the title, abstract, and the introduction.
  • The manuscript reads as a loose mixture of rather arbitrary excerpts from mitochondrial structural biology intercalated with clinical phenotypes of select syndromes. Sections of the manuscript appear to be disjointed due to the lack of mechanistic connections between etiology of diseases and their molecular underpinnings. In their current form consideration of molecular mechanisms appear too superficial for the scope of IJMS while descriptions of syndromes are too heavy on the clinical side to bridge the two.
  • The content of this manuscript is imbalanced with nearly half of the citation appearing in the last three pages. At the same time, significant part of the first half in essence recaps textbook knowledge with little or no citations.
  • This is compounded by confusing, unconventional, or mistaken terminology often used by the authors: ΔP instead of Δμ for chemiosmotic potential, decomposition of rhodopsin, flow of current, electric vs electronic, direct vs threshold vs graded response, ROS as by-product, 8993T>A, hydrolytic ATPases vs oligomycin-sensitive ATP synthase encoded by ATPase6, redundant sentences, misplaced and repeated definition of abbreviations, and many more issues.
  • I recommend to structure manuscript more clearly using sections and subsection, which will help to avoid shallow passages such as sections 2 and 8.

Author Response

We thank the Reviewer for underscoring some editing problems, essentially due to the difficulty to communicate assiduously because of the pandemic (Padova and Bologna are 130 km apart from each other). Now these mistakes, including the correct attribution of the figure temporarily denominated XY has been completed. All figures are now cited in the text and some legends have been changed. The referral to the text was meant to prevent the legends to become too heavy. The term Mitochondrial Retinopathies was the topic originally proposed by the Editor, and on the other hand, mitochondrial pathogenetic roles can be found in almost every pathological condition nowadays so we decided to stick to retinopathies associated with ascertained primary mitochondrial disorders. This is now explicitly said at the beginning of the paper. We have however mentioned the possible association with MS and with age-related maculopathy. The association with diabetes is certainly interesting but I do not think that we can define the diabetic retinopathy a mitochondrial retinopathy, so in this case we have not addressed this topic. The overall evaluatioon that the referee is giving about our work makes it virtually not amendable but we disagree with this evaluation. The review is organized in an introductory part explaining the essential biology and genetic of mitochondria, followed by a survey on the physiology bof the retina, with special attention to the aspects related to mitochondrial pathophysiology. We then discuss the two main entities of mitochondrial retinopathy, i.e. retinal degeneration or retinitis pigmentosa, addressing the two most important entitoes, NARP/MILS and Kerns Sayre syndrome; and degeneration of RGCs and their axons, associated with LHON mutations, OPA1 mutations, and a number of additional OPA-associated conditions due to several nuclear genes whose mutation can determine either non-syndromic or syndromic optic atrophy, or other possible syndromes. We also mention the recent discovery of autosomal recessive LHON, in addition to the traditional mtDNA-associated mutations. As suggested by referee 2, the review terminates with a chapter on the most recent advances in therapy, especially for LHON, and a succinct chapter of concluding remarks. Frankly I don't see how the referee can evaluate this scheme as highly disorganized and scrambled and unbalanced. Concerning the terminology, whilst deltaµ is currently not used, deltap designates the proton-motive force (not by me but by Peter Mitchell in his BBA paper in 1966 now reported as reference). To reconcile with the opinion of referee 1 mitochondrial membrane potential was denominated MtMP, a term widely used in the current literature, whereas the proton-motive force was specifically tered as delta-p according to Mitchell's nomenclature. The H+ ATPsynthase, so denominated throughout the manuscript, is currently designated also as mitochondrial ATPase, since it can work in reverse, hydrolizing ATP in ADP and Pi, which is the way it is currently measured spectrophotometrically. In addition somne subunits of the enzyme, for instance subunit A dorming the proton channel impaired in NARP/MILS, are encoded by genes denominated ATPase followed by a number, for instance ATPase6 for subunit A, ATPase8 for the other subunit encoded by mtDNA. We feel that this is a very marginal problem anyway. Other terms criticised by Referee 1 are currently and consistently present in the literature on mitochondrial biology; we are talking about electronic current (it can be changed into electronic flow but it is the same) whereas electric current refers to action potentials traveling through the axons of retinal neurons. ROS are considered universally as an obligatory by product of respiration. Likewise, the term graded is used in the Guyton's textbook of Physiology we consulted to designated the nature of electrotonic conduction in most of the retinal synapses. Although we shortened this introductory chapter on retinal physiology, as suggested by referee 2, we think that the IJMS readership may not know or may have forgotten some concepts that we think essential to understand the relevance of the bioenergetic failure occurring in the retina in the disorders we are illustrating ion the subsequent sections of the paper. Altogether, we tried to amend as much as possible the manuscript and legends according to what we agree of the objections of the referee1. We are sorry that he or she did not like the work, and we shall ask the Editor if we should wthdraw it atogether , due to the severe criticisms of this referee. 

Reviewer 2 Report

Zeviani et al has compiled a elaborate review on mitochondrial retinopathies. This review will serve as a current update on mitochondrial retinal diseases and will re-emphasize the importance of mitochondrial diseases and their impact on Ophthalmic diseases. The review is well written and touched the basics of mitochondrial metabolism and retinal physiology. However, the following points may improve the quality of the review article and will add more interest to the readers.

  1. Authors may consider reducing the description of retinal physiology which include physiology of photoreceptors and the RPE as the topic here is to educate the readers mainly on the impact of mitochondrial genetic defect on retinal physiology. Authors can explain the impact of these genetic changes on each physiological function such as respiratory chain dysfunction, DNA instability, mitochondrial dynamics and quality control.
  2. Authors should include a table with all the mitochondrial mutations identified so far to cause retinal dysfunction and the table must include type of the disease and which function in the mitochondria is affected.
  3. Authors should include therapeutic opportunities as a subsection which will be interest and useful for the readers. which should include all the treatment interventions so far to treat mitochondrial retinopathies.

Author Response

We thank Referee 2 for the pro-active suggestions to improve the manuscript. We have shorten by almost three pages the introduction on mitochondrial and retinal biology. Nevertheless, we think that the IJMS readership may not know in depth either mitochondrial biochemistry and genetics, or retinal organization and physiology. Since we do not know whether the editorial plan will contain an exhaustive introductory chapter explaining these concepts, we think opportune to leave this introductory part, albeit shortened, to let the interested reade kave the conceptual tools to better understand the subsequent sections. Upon suggestion of Referee 2 we added a Table I which summarizes the main genes/mutations and associated sretinal and extra-retinal clinical signs of the associated disease. Thank you for the suggestion. We also addede a final chapter on the current trials and advanced therapeutic approaches, especially concerning LHON. Thank you again for this useful suggestion.